# Effects and Mechanisms Activated by Treatment with Cationic, Anionic and Zwitterionic Liposomes on an In Vitro Model of Porcine Pre-Pubertal Sertoli Cells

**DOI:** 10.3390/ijms24021201

**Published:** 2023-01-07

**Authors:** Giulia Collodel, Elena Moretti, Daria Noto, Roberta Corsaro, Cinzia Signorini, Claudia Bonechi, Lorenzo Cangeloni, Giovanni Luca, Iva Arato, Francesca Mancuso

**Affiliations:** 1Department of Molecular and Developmental Medicine, University of Siena, 53100 Siena, Italy; 2Department of Biotechnology, Chemistry and Pharmacy, University of Siena, 53100 Siena, Italy; 3Department of Experimental Medicine, University of Perugia, 06100 Perugia, Italy; 4Division of Medical Andrology and Endocrinology of Reproduction, Saint Mary Hospital, 05100 Terni, Italy

**Keywords:** apoptosis, actin and vimentin filaments, immunofluorescence, in vitro study, male infertility, Sertoli cells, TEM, zwitterionic liposomes

## Abstract

Liposomes have been successfully used as drug-delivery vehicles, but there are no clinical studies on improved fertility and the few reported experimental studies have been performed in animal models far from humans. The aim of this paper was to study the effects of treatment with cationic, anionic and zwitterionic liposomes on our superior mammalian model of porcine prepubertal Sertoli cells (SCs) to find a carrier of in vitro test drugs for SCs. Porcine pre-pubertal SCs cultures were incubated with different liposomes. Viability, apoptosis/necrosis status (Annexin-V/Propidium iodide assay), immunolocalisation of β-actin, vimentin, the phosphorylated form of AMP-activated protein Kinase (AMPK)α and cell ultrastructure (Transmission Electron Microscopy, TEM) were analysed. Zwitterionic liposomes did not determine changes in the cell cytoplasm. The incubation with anionic and cationic liposomes modified the distribution of actin and vimentin filaments and increased the levels of the phosphorylated form of AMPKα. The Annexin/Propidium Iodide assay suggested an increase in apoptosis. TEM analysis highlighted a cytoplasmic vacuolisation. In conclusion, these preliminary data indicated that zwitterionic liposomes were the best carrier to use in an in vitro study of SCs to understand the effects of molecules or drugs that could have a clinical application in the treatment of certain forms of male infertility.

## 1. Introduction

Sertoli cells (SCs) crosstalk with germ cells including spermatogonia, spermatocytes, haploid spermatids and spermatozoa during spermatogenesis. SCs provide nutrients, paracrine factors and cytokines and facilitate the generation of several biologically active peptides, which include F5-, non-collagenous 1 (NC1)- and laminin globular (LG)3/4/5-peptide, to modulate cellular events across the epithelium [1]. In normal testis, the blood testis barrier/SC barrier protects most of the auto-antigenic germ cells by limiting access by the immune system and sequestering these ‘new antigens’. SCs also modulate testis immune cells (induce regulatory immune cells) by expressing several immunoregulatory factors, thereby creating a local tolerogenic environment that is optimal for the survival of non-sequestered auto-antigenic germ cells [2]. In the testis, follicle-stimulating hormone (FSH) controls the function of SCs through FSH receptors (FSH-r), which are only present in SCs. A functioning testicular microenvironment is critical for normal spermatogenesis and reproductive function. Alterations in the testicular microenvironment can lead to impaired fertility due to a loss of germ cell support [3,4]. Further understanding of the unique characteristics of the human testicular microenvironment (organisation, cell types, integration of signalling cues) will therefore be critical to finally achieving full human spermatogenesis in vitro. Primary cell cultures are considered the “gold standard” of in vitro models and they can more closely mimic the real characteristics of target tissues in vivo [5]. 

The successful isolation and culture of Sertoli cells depend on a series of delicate processes of mechanical isolation and enzymatic digestion of the testicular tissue. Rat and mouse Sertoli cells are obtained by a similar protocol, whereas bovine and human Sertoli cells require a more extensive mechanical and enzymatic processing [6]. In particular, porcine prepubertal SCs is an experimental animal model that exhibits significant physiological similarity with humans, meaning that the results obtained using this approach, from the perspective of translational medicine, can be applied to humans. Porcine prepubertal SCs have been recognised as a valuable in vitro system to study the effects of heavy metals, toxic substances [7,8,9] and drugs [10]. In addition, they were used to test the protective action of antioxidant substances under heavy metal exposure [11] and to identify signalling pathways [12]. Microencapsulated porcine pre-pubertal SCs have been employed for cell therapy in pre-clinical studies for several chronic/immune diseases [13]. 

Liposomes are mainly composed of phospholipids and could be categorised differently according to their structure and size [14]. They can be adsorbed into the membrane of cells specifically or non-specifically. The mechanism and degree of liposome–cell interaction is strongly influenced by the nature and density of the charge on the liposome surface. The liposomes can include charged chemical groups that confer an overall zwitterionic, positive or negative charge. In an in vitro study, the liposome surface charge influences many activities such as binding and endocytosis. Cationic liposomes induced apoptosis in mouse splenic macrophages and macrophage-like cells [15], whereas anionic liposomes have been fairly restricted to the delivery of specific therapeutic macromolecules, [16] influencing many aggregation or absorption phenomena [17]. Encapsulation efficiency, which refers to the amount of drug contained in liposomes compared to the total amount of active ingredient, is one of the most important parameters for liposome formulation because it directly reflects the concentration of the drug. The factors affecting the encapsulation efficiency depend on the properties of the liposome and the encapsulated drug. Therefore, EE% indicates the amount of active ingredient that can actually be transported to the active site and released.

Bonechi et al. [18] set up the synthesis of anionic, cationic and zwitterionic liposomes, loaded with quercetin or rutin. The liposome encapsulation of antioxidants makes their internalisation by NIH3T3 fibroblasts easier. Moreover, zwitterionic and anionic liposomes loaded with rutin protect cells via oxidative stress.

One of the most promising applications is the use of liposomes as drug-delivery vehicles of a vast range of different drugs, including anti-cancer chemotherapeutics [19,20], fungicides [21], hormones [22], enzymes [23] and genetic material [24], among others.

At present, there are no clinical studies on the improvement of male fertility by using liposomes as carriers of drugs or supplements and the few reported experimental studies have been performed in animal models far from humans, although with promising results. In fact, the motility of bovine spermatozoa, treated with vitamin E liposome preparations, was significantly improved after semen cryopreservation [25]. Recently, it was observed that the nasal delivery of NGF pre-encapsulated with liposomes restored spermatogenesis in an azoospermic mice model [26].

Since SCs represent the real director of spermatogenesis [27], the present research focused on the effects and mechanisms activated by treatment with cationic, anionic and zwitterionic charged liposomes on our superior mammal model of SCs, to find a carrier of in vitro test drugs for SCs.

## 2. Results

### 2.1. Sertoli Cell Isolation and Characterisation

SCs were isolated, characterised by immunofluorescence and cytofluorimetric analysis and examined in terms of viability and function, namely the production of AMH and inhibin B. The purity of isolated SCs was about 95% with negligible contaminating cells [13]. 

### 2.2. Liposome Size and Surface Charge

Physical characteristics of the liposomes, i.e., size, polydispersity index (PDI) and ζ-potential, were determined and reported in Table 1. The PDI values suggest that all liposomes, empty and loaded with rhodamine, were monodisperse. Since the aggregation may be used as a measure of liposome physical stability, we can conclude that the incorporation of rhodamine in all different liposome bilayers does not modify their physical stability.

The mean diameters of liposomes were in accord with extrusion processes and the insertion of rhodamine does not change the chemical behaviour of phospholipids in the double layers. 

By analysing the data reported in Table 1 we can observe that empty DOPC/DOPE (zwitterionic) liposomes had a negative ζ-potential, though the net polar head charge of zwitterionic phospholipids was zero. The insertion of rhodamine does not change the surface charge of liposome systems, as indicated by ζ-potential values.

### 2.3. Encapsulation Efficiency (EE) 

To study the effect of the loading on the activity of rhodamine, UV-visible analysis was carried out after the encapsulation step (as reported in the Section 4). This experimental data is essential to calculate the EE% (Table 2). Good EE% values were obtained for all formulations. We observed non-significant differences in the EE% of rhodamine between zwitterionic, cationic and anionic liposomes. All synthesised liposomes have a good ability to include rhodamines.

### 2.4. SC Incubation with Liposomes Loaded with Rhodamine

Cationic, anionic and zwitterionic liposomes loaded with rhodamine (diluted 1:10,000) were incubated with SCs for 6, 12 and 24 h. Untreated SCs were used as a control. In all of the analysed samples previously incubated with liposomes and loaded with rhodamine, SCs appeared red, the colour of rhodamine, indicating a very high level of adhesion or fusion. We observed that the cells were stained, and normal cell morphology was maintained after 6, 12 and 24 h incubation. Despite SCs treated with zwitterionic rhodamine-loaded liposomes showing a similar density compared to untreated samples after 12 and 24 h, those incubated with anionic and cationic rhodamine-loaded liposomes showed a lower density, suggesting a cytotoxicity activity of these liposomes.

For this reason, the experiments were performed by incubating cells for 6 h with the three types of rhodamine-loaded liposomes. SC morphology and density were similar in all treated samples as well as in untreated cells, indicating that all of the liposomes used were non-toxic.

Figure 1 highlights the cultured SCs after 6 h incubation with all the types of liposomes; rhodamine stain was evident in all treated samples, suggesting that liposomes interact with porcine SCs.

Then, all of the following experiments were conducted at 6 h incubation using empty liposomes (cationic, anionic and zwitterionic) and different cell endpoints were evaluated, as well as cell viability, morphology, the percentage of apoptosis, the phosphorylated form of AMPK and the organisation of β-actin and vimentin filaments. Three experiments for each endpoint were carried out. 

Cell viability ranged from 89 to 97% (Table 3) and the data were similar. However, the number of total cells in samples incubated with anionic and cationic liposomes was slightly lower, but not significantly, compared to that of the SC control. 

### 2.5. Immunofluorescence 

Immunofluorescence analysis showed that the localisation of β-actin, vimentin and phospho-AMPKα in SCs was different in samples incubated with cationic, anionic and zwitterionic liposomes. 

The distribution of β-actin appeared uniform from the nucleus to the plasma membrane (Figure 2a) after incubation with zwitterionic liposomes. (Figure 2a) showed that SCs incubated with cationic and anionic liposomes highlighted very intense spots in the cytoplasm at a percentage of 35% and 25% of examined cells, respectively (Figure 2b). 

The normal localisation of vimentin appeared in the entire cytoplasm and close to the nucleus (Figure 2c); however, after incubation with cationic and anionic liposomes, a percentage of 42% and 37% of cells, respectively, lost the uniform cytoplasmic distribution of vimentin, displaying strong pointed signals in the cytoplasm (Figure 2d). This type of localisation was not present in SCs from control and from zwitterionic liposome samples.

The Phospho-AMPKα labelling was brighter in SCs treated with cationic and anionic liposomes that showed the presence of numerous spots in the cytoplasm (Figure 2e, 40% of the cells) compared to controls and cells incubated with zwitterionic liposomes (Figure 2f). 

The Annexin-V and Propidium Iodide (AnV/PI) assay showed that a percentage of 85–90% of SCs was viable (annexin V negative/PI negative) in untreated cells (control) or cells treated with zwitterionic liposomes; the apoptotic green signal was detected in only 5% of cells. On the contrary, more than 40% of SCs treated with anionic or cationic liposomes highlighted green spots (early apoptotic cells, annexin V positive/PI negative, Figure 3a). Red spots (apoptotic/necrotic cells, annexin V positive/PI positive or annexin V negative/PI positive) were reduced in all the samples (2–3%, Figure 3b).

### 2.6. Ultrastructural Analysis

According to TEM analysis, the controls and SCs treated with zwitterionic liposome SCs (Figure 4a,b) showed a normal ultrastructure: a regular chromatin condensation and well-organised cytoplasm were observed. In cells treated with anionic and cationic liposomes, TEM examination allows the detection of an increased percentage of cells (41.33% ±  2.50%) with apoptotic features such as marginated chromatin and vacuoles (Figure 4c,d, more than one third of sections) compared to untreated ones (7.5%  ±  1.05%). 

## 3. Discussion

In this paper, we evaluated the effect of zwitterionic, cationic and anionic liposomes on primary cultures of neonatal porcine SCs. 

Since the SCs interact directly with germ cells and are vital to providing morphological and nutritional support for spermatogenesis, their dysfunction is often the cause of spermatogenic failure [28]. The cultures of SCs are fundamental for studies of the blood-testis barrier and testicular immune privilege, as well as the analysis of spermatogenesis-associated events both in vitro and in vivo. The most common methods of isolation of SCs and studies in this field were applied in animal [29] and human models [30].

Our superior mammal model of porcine prepubertal SCs represents an opportunity to evaluate the effects of treatments on SCs that could find applications in male infertility therapy. In a previous study, zwitterionic and anionic liposomes were loaded with rutin protected fibroblasts by oxidative stress. Liposome stability together with their good in vitro cytocompatibility, both empty and loaded with antioxidant molecules, makes these systems suitable candidates for drug delivery systems [18]. It is known that liposomes have magnetic properties and the ability to penetrate cell membranes and deliver their loads into cell cytoplasm [31]; these characteristics make liposomes a useful tool in several clinical applications [32,33].

Using rhodamine-loaded liposomes, we observed that all the tested liposomes interacted with the cells at 6, 12 and 24 h of incubation without any evident toxicity. Moreover, all experimental chemical data suggest that the liposomes studied in this paper (zwitterionic, anionic and cationic) are stable and are able to include rhodamine, as previously demonstrated in other experiments [18].

After the demonstration that liposomes can interact with SCs, we tested the effect of empty liposomes (cationic, anionic and zwitterionic) after 6 h incubation to choose the better liposome that could be used as a carrier of molecules in SCs. We decided on a 6 h incubation due to our observations and because an external stimulus in the form of polymeric microparticles and lipopolysaccharides was able to activate pathways involved in cell phenotype modification within 5 h in a previous study [12]. 

In the present study, we assessed the effects of different liposomes on SC functional competence through the evaluation of AMH and inhibin B secretion, as they are specific and important markers of SC functionality [8]. At first analysis, the three liposomes appeared similar in their behavior since SCs retained a similar percentage of viability after 6 h incubation. However, further analysis revealed that incubation with anionic and cationic liposomes may determine changes in cell density and cytoplasm organization. Indeed, the distribution of actin and vimentin filaments changed in the presence of anionic and cationic liposomes and the levels of the phosphorylated form of AMPKα increased. In addition, the Annexin/Propidium Iodide assay suggested an increase in apoptosis. The actin and vimentin alterations and the plausible increase in apoptosis were confirmed by TEM analysis, which allowed the presence of cytoplasmic vacuolisation to be highlighted. It is known that the presence of high vacuolisation is a typical feature of apoptotic process and negatively influences the filamentous organisation in other cell models [34,35]. A gradual disappearance of vimentin concomitant with the presence of vacuoles in SC cultures from 21-day-old C57Bl/6N mice treated with mono(2-ethylhexyl) phthalate has been described.

Different authors reported that cationic liposomes induce apoptosis in the macrophage-like cell line RAW264.7 [36,37,38]. In particular, Aramaki et al. [36] have shown that the cationic liposome-induced mitochondrial membrane depolarisation and the release of cytochrome c from mitochondria triggers an apoptotic processe. Cationic liposomes composed of stearylamine induced apoptosis in macrophages by involving the proteoglycan-actin cytoskeleton-reactive oxygen species (ROS) generation pathway [39]. 

We have also evaluated AMPK, an important energy sensor that increases different metabolic pathways. Alterations in AMPK signalling decrease mitochondrial biogenesis, increase cellular stress and induce inflammation, which are typical events of the aging process and have been associated with several pathological processes. 

It has been reported that the phosphorylation and activation of AMPK cause cell cycle arrest in several cell lines; in particular, its activation reduced the proliferation, stimulated by FSH, of the SCs [40]. The data of the present research suggest that an increased phospho-AMPK signal may be related to the change in proliferative pathways in the presence of cationic and anionic liposomes. Both of these liposomes negatively influenced cell density. Moreover, the increase of phospho-AMPKα in the cytoplasm of SCs could be a result of apoptosis stimulated through p38 MAP kinase-caspase-8-Bid pathway as reported in macrophage-like RAW264.7 cells treated with cationic liposomes [38]. 

All of these data indicated that the anionic and cationic charged liposomes might induce apoptosis and zwitterionic liposomes are the best choice for SC applications. Previously, our group tested the effect of different liposomes on sperm motility and viability and only DOPC/DOPE liposomes did not affect the studied parameters, confirming the expectation of zwitterionic liposomes [41].

It is known that liposomes have improved the therapeutic efficacy of drugs through stabilising compounds, overcoming obstacles to cellular and tissue uptake and increasing drug biodistribution to target sites in vivo, while minimising systemic toxicity [33]. 

## 4. Materials and Methods

### 4.1. Primary Cultures of Neonatal Porcine Sertoli Cells

Animal studies were conducted in agreement with the guidelines adopted by the Italian Approved Animal Welfare Assurance (A-3143-01) and European Communities Council Directive of 24 November 1986 (86/609/EEC). The experimental protocols were approved by the University of Perugia. Number 2 Danish Duroc neonatal pigs (15 to 20 days old) underwent bilateral orchidectomy after general anaesthesia with ketamine (Ketavet 100; Intervet, Milan, Italy), at a dose of 40 mg/kg and dexmedetomidine (Dexdomitor, Orion Corporation, Finland), at a dose of 40 g/kg and were used as SC donors. Specifically, pure porcine neonatal SC were isolated, characterised and tested for functional competence according to previously established methods [42,43].

### 4.2. Anti-Mullerian Hormone (AMH) and Inhibin B Determination

Aliquots of the culture media from all the experimental groups were collected and stored at −20 °C for the assessment of AMH (AMH Gen II ELISA, Beckman Coulter; intraassay CV = 3.89%; interassay CV = 5.77%) and inhibin B (inhibin B Gen II ELISA, Beckman Coulter, Webster, TX, USA; intraassay CV = 2.81%; interassay CV = 4.33%) secretion, as previously described [7].

### 4.3. Liposome Preparation 

DOPC (1,2-dioleoyl-sn-glycero-3-phosphocholine), DOPE (1,2-di-(9Z-octadecenoyl)-sn-glycero-3-phosphoethanolamine), DOPA (1,2-dioleoyl-sn-glycero-3-phosphate) and DOTAP (1,2-dioleoyl-3-trimethylammonium-propane) were purchased from Avanti Polar Lipids Inc., Alabaster, AL, USA.

### 4.4. Liposome Preparation: Cationic, Anionic and Zwitterionic 

DOPC/DOPE, DOPE/DOPA and DOTAP/DOPE liposomes were prepared at 1:0.5 molar ratio with a total lipid concentration of 1.0 × 10^−2^ M. 

Liposomes were prepared in a round bottom vial by mixing the appropriate amounts of stock solutions, which were 4 × 10^−2^ M in chloroform for lipids. 

Two different sets of cationic, anionic and zwitterionic liposomes were considered: rhodamine-loaded and empty liposomes.

A dry lipid film was obtained by evaporating the solvent under vacuum overnight. Rehydrating with Milli-Q grade H_2_O yielded a multilamellar dispersion. In the loaded liposomes, the rhodamine solution was used to hydrate the lipidic film, obtaining a final molar ratio of 1:0.5 (total lipids: rhodamine). Multilamellar vesicles were obtained by vortexing, which were then submitted to eight freeze/thaw cycles. 

This method improved the homogeneity of the size distribution in the final suspension. Liposomes were then reduced in size and converted to unilamellar vesicles by extrusion through 100 nm polycarbonate membranes. Twenty-seven extrusions were performed with the LiposoFast apparatus (Avestin, Ottawa, Canada). All liposomes were stored at 4 °C.

### 4.5. Size and Surface Charge of Liposomes

The size and surface charge of the empty, functional loaded liposomes were measured by Dynamic Light Scattering, DLS, (Coulter Sub-Micron Particle Analyzer N4SD, equipped with a 4 mW helium-neon laser and 90 u detector) and Zeta potential (Coulter DELSA 440 SX), respectively.

Liposome dispersions were diluted with Tyrode Buffer Saline pH 7.40 and ζ-potential values were measured at 25 °C. As the radii of liposomes were always large enough compared with the Debye–Huckel parameters, the ζ-potentials were calculated directly by means of the Helmoholtz–Smolowkovski equation (by the zetasizer) [44].

Size was also calculated in the same experiments, according to the procedure described by Langley [45].

### 4.6. Encapsulation Efficiency (EE) of Rhodamine

The EE% data was obtained by UV-VISIBLE spectra, recorded at 25 °C with a Perkin-Elmer Lamda 25 spectrophotometer (10 mm cuvettes). Before spectra recording, liposome disruption was carried out to dispose of the scattering background (scaling as λ-4), due to large aggregates in solution, which can affect precise intensity evaluation. To disrupt liposomes and release the entrapped rhodamine, samples underwent several cycles of freezing (−32 °C). A calibration curve was built by measuring the absorbance of solutions with known rhodamine concentration at 530 nm.

Encapsulation efficiency (EE%) describes the amount of drug encapsulated in the liposome relative to the total amount of drug used in liposomal synthesis. The EE% can be calculated using the following equation:EE% = E_drug_/T_drug_ × 100
where T_drug_ is the total amount of drug added during synthesis and E_drug_ is the experimentally determined amount of drug.

### 4.7. Primary Cultures of Neonatal Porcine Sertoli Cells and Treatments

To evaluate the potential interaction of liposome-cells and to choose the timing of incubation, SCs were first incubated with cationic, anionic and zwitterionic liposomes loaded with rhodamine. 

Briefly, unexposed and exposed SCs monolayers were grown on cover glasses (12 mm, Thermo Scientific, Braunschweig, Germany), coated with 0.1% DIFCO gelatin (Thermo Scientific, Braunschweig, Germany) and fixed in 4% paraformaldehyde (pH 7.4, Thermo Scientific, Braunschweig, Germany) for 15 min. 

The cells were washed in PBS three times (5 min each time), then incubated with cationic, anionic and zwitterionic liposomes loaded with rhodamine diluted 1:10,000 for 6, 12 and 24 h and counterstained for 1 min with DAPI (Sigma-Aldrich Co., St. Louis, MO, USA). The cells were mounted with ProLong^®^ Gold antifade reagent (Molecular Probes, NY, USA) and analysed using a BX-41 microscope (Olympus, Tokyo, Japan) equipped with a fluorescence photo camera.

For successive experiments, SC cultures were incubated for 6 h with cationic, anionic and zwitterionic empty liposomes. The culture medium was recovered and stored at −80 °C, while cells were collected to evaluate viability (Automated Cell Counter, Invitrogen, CA, USA), apoptosis/necrosis status (Annexin-V/Propidium iodide assay) and cell ultrastructure (Transmission Electron Microscopy, TEM) and stored at −20 °C until use for immunofluorescence.

### 4.8. Transmission Electron Microscopy (TEM)

SCs were fixed in Karnovsky fixative at 4 °C for 2 h and washed in 0.1 M cacodylate buffer (pH 7.2) for 12 h. Then, cells were postfixed in 1% buffered osmium tetroxide for 1 h at 4 °C, dehydrated in a graded ethanol series (50%, 75%, 95% and 100%) and embedded in Epon Araldite. Ultrathin sections were obtained with the Supernova Ultramicrotome (Reichert Jung, Vienna, Austria), mounted on copper grids and stained with uranyl acetate and lead citrate. Stained grids were observed with a Philips CM12 TEM (Philips Scientific, Eindhoven, the Netherlands; Centro di Microscopie Elettroniche “Laura Bonzi”, ICCOM, Consiglio Nazionale delle Ricerche (CNR), Via Madonna del Piano, 10, Firenze, Italy). Three experiments were performed. For each sample, at least 100 cell sections were evaluated. 

### 4.9. Immunofluorescence Analysis

The immunolocalisation of β-actin, vimentin and the phosphorylated form of AMPKα was performed in SCs treated with cationic, anionic and zwitterionic liposomes and untreated cells. After thawing, coverslips were rehydrated in PBS and treated with a blocking solution (PBS-BSA 1% NGS 5%) for 20 min, before being incubated overnight at 4 °C with primary antibodies anti-β-actin (Santa Cruz Biotechnology, Dallas, USA) diluted 1:100, anti-vimentin (Santa Cruz Biotechnology, Dallas, USA) diluted 1:200 and anti-Phospho-AMPKα (Thr172) (Cell Signaling Technology, Danvers, MA, USA) diluted 1:100. The reaction was revealed by a goat anti-mouse IgG-Alexa Fluor 568 (Invitrogen, Thermo Fisher Scientific, Carlsbad, CA, USA) diluted 1:100 and an anti-rabbit antibody raised in a goat Alexa Fluor^®^ 488 conjugate (Invitrogen, Thermo Fisher Scientific, Carlsbad, CA, USA), diluted at 1:100 1h at room temperature. Incubation without the primary antibodies was used as control. Nuclei were stained with 4,6-diamidino-2-phenylindole (DAPI) solution (Vysis, Downers Grove, IL, USA). Finally, the slides were mounted with 1,4-diazabicyclo [2.2.2]octane (DABCO, Sigma-Aldrich, Milan, Italy). Observations were made with a Leica DMI 6000 Fluorescence Microscope (Leica Microsystems, Wetzlar, Germany) and the images were acquired by the Leica AF6500 Integrated System for Imaging and Analysis (Leica Microsystems, Wetzlar, Germany). The experiments were performed on three samples.

### 4.10. Annexin-V and Propidium Iodide Labelling

All procedures were carried out as in previous investigations of cultured cells [46]. For the assessment of apoptosis and necrosis, SCs treated with cationic, anionic and zwitterionic liposomes and control (untreated) were labelled using the Vybrant Apoptosis Assay kit (Invitrogen Ltd., Renfrew, UK) according to the manufacturer’s instructions. After treatments, controls and treated cells were harvested, washed with PBS, centrifuged at 400 g for 5 min and resuspended with Annexin binding buffer (ABB) to obtain a concentration of 1 × 106 cells/mL. For each cell suspension, 5µL of Annexin-V conjugated to fluorescein isothiocyanate dye (AnV-FITC) and 1 µL of Propidium iodide (PI) working solution (100 mg/mL) were added and incubated for 15 min at room temperature. After a careful washing with ABB, a drop of the cell suspension was smeared on each glass slide. Slides were mounted with DABCO (Sigma-Aldrich, Milan, Italy). Observations were made with a Leica DMI 6000 Fluorescence Microscope (Leica Microsystems, Wetzlar, Germany) and the images were acquired by the Leica AF6500 Integrated System for Imaging and Analysis (Leica Microsystems, Wetzlar, Germany). One hundred cells from each sample were examined. Green spots represent early apoptotic cells (Annexin V positive/PI negative), while red spots indicate apoptotic/necrotic cells (Annexin V positive/PI positive or Annexin V negative/PI positive). All experiments were carried out in triplicate. The results are expressed as the percentage of cells that were apoptotic and necrotic.

### 4.11. Statistical Analysis

Statistical analysis was accomplished with the SPSS version 17.0 for Windows software package (SPSS Inc, Chicago, IL, USA). The Kruskal-Wallis test was used to compare the difference among different groups and then Mann-Whitney test was applied to determine which groups differed statistically from each other. Data were reported as mean and standard deviation. *p* < 0.05 was considered significant.

## 5. Conclusions

In conclusion, the preliminary data indicate that zwitterionic liposomes are the best carrier to use in an in vitro study on SCs to understand the effects of molecules or drugs that could be used in the treatment of certain forms of male infertility.

Once again, it is important to emphasise that the choice of this type of liposome has been carried out using different techniques such as immunocytochemistry and TEM, allowing us to better understand cell conditions that are not always evident with only biochemical or molecular analyses.

## Figures and Tables

**Figure 1 ijms-24-01201-f001:**
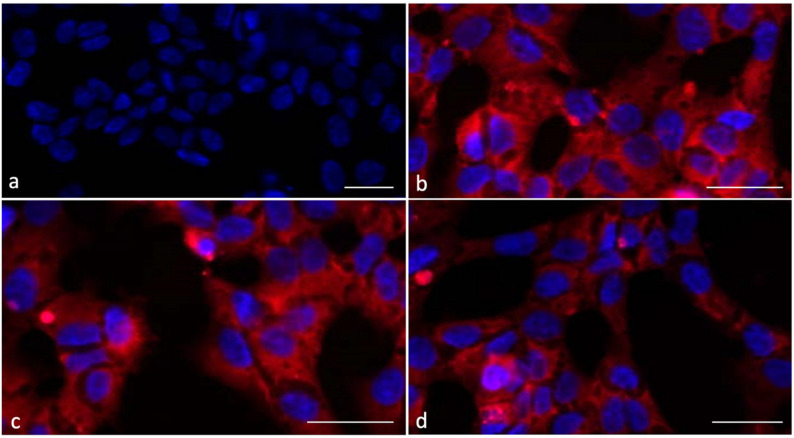
UV micrographs of SCs after 6 h incubation: untreated (**a**) and treated with cationic (**b**), anionic (**c**) and zwitterionic (**d**) rhodamine-loaded liposomes. The staining is present in the cytoplasm of all the examined samples. Magnification 100×. Bar: 20 µm.

**Figure 2 ijms-24-01201-f002:**
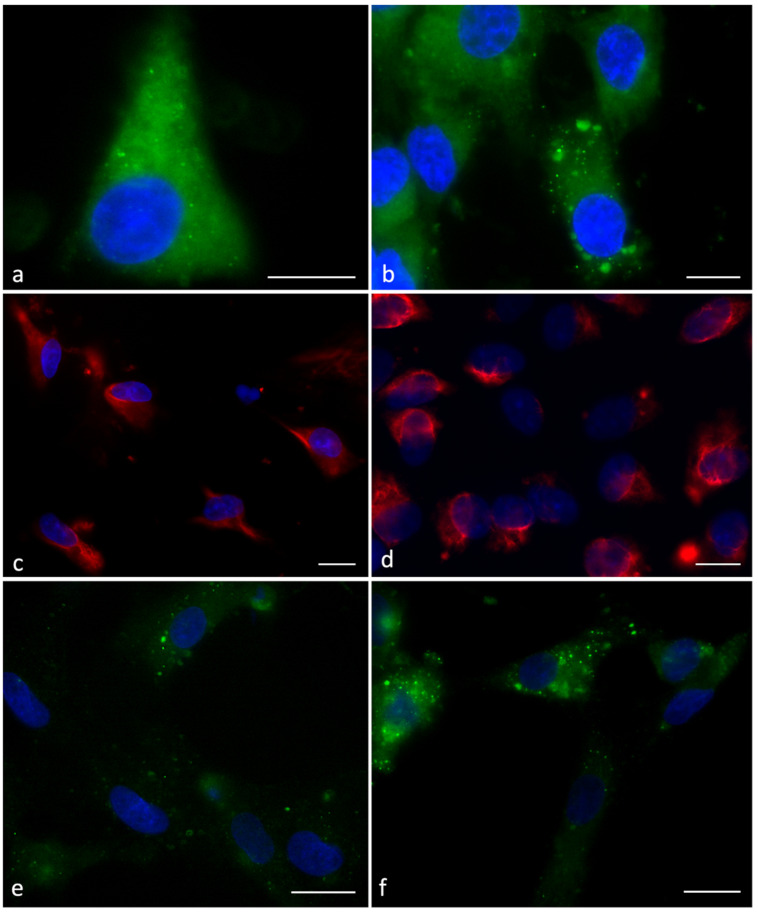
UV micrographs of SCs after 6 h incubation with zwitterionic and cationic liposomes. (**a**,**b**) β-actin. SCs treated with zwitterionic liposomes (**a**), a homogeneous distribution is present in the cytoplasm similar to that detected in controls; SCs treated with cationic showed intense spots in the cytoplasm (**b**). (**c**,**d**) Vimentin. SCs treated with zwitterionic liposomes showed uniform distribution (**c**); SCs treated with cationic liposomes showed a staining strongly pointed (**c**) as detected in the presence of anionic liposomes. (**e**,**f**) Phospho-AMPKα. The Phospho-AMPKα labelling was lower after incubation with zwitterionic liposomes (**e**) and brighter in SCs treated with cationic liposomes (**f**). Bar 10 µm.

**Figure 3 ijms-24-01201-f003:**
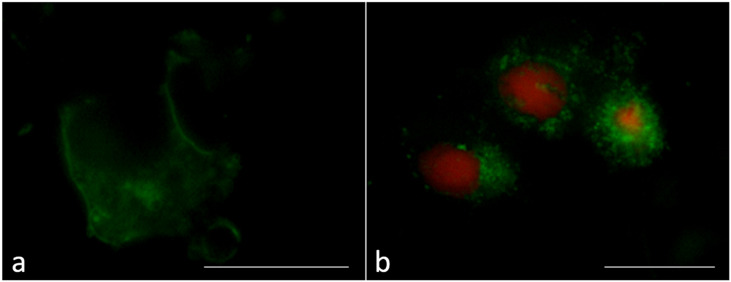
Annexin-V/Propidium Iodide assay in the Sertoli cells (SCs). Apoptotic (green, (**a**)), necrotic (red, (**b**)) and apoptotic/necrotic cells (green/red, (**b**)) are shown. Bar 20 μm.

**Figure 4 ijms-24-01201-f004:**
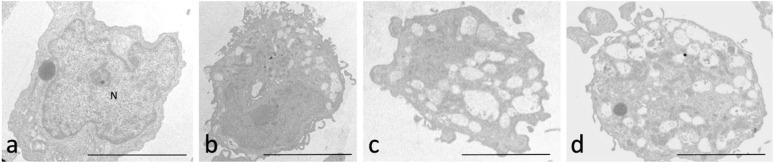
Transmission electron microscopy micrographs of SCs. In panel (**a**) a SC cell treated with zwitterionic liposomes is shown: the cell has a regular nucleus (N) and organised cytoplasm. The very small percentage (5%) of cells highlights a partial vacuolisation (**b**). In panel c and d, SCs are displayed after incubation with anionic (**c**) and cationic (**d**) liposomes. In a high percentage of cells (41%), the cytoplasm appears rich in vacuoles and the presence of organelles is reduced. The plasma membrane is the integer in all the samples. Bar 12 μm.

**Table 1 ijms-24-01201-t001:** Size and surface charge of empty liposomes and rhodamine-loaded liposomes obtained by extrusion through 100 nm polycarbonate membranes. DOPC: 1,2-dioleoyl-sn-glycero-3-phosphocholine; DOPE: 1,2-di-(9Z-octadecenoyl)-sn-glycero-3-phosphoethanolamine.

Liposome Composition	Mean Diameter (nm) ± SD	P.D.I.	ζ-Potential (mV) ± SD
DOPC/DOPE (1:0.5)	125.9 ± 15.4	0.12	−18.27 ± 3.72
DOPC/DOPE + rhodamine	159.8 ± 10.4	0.18	−17.93 ± 2.41
DOPC/DOTAP (1:0.5)	139.0 ±16.2	0.08	60.12 ± 3.67
DOPC/DOTAP + rhodamine	165.2 ± 9.4	0.16	52.34 ± 4.65
DOPE/DOPA (1:0.5)	149.1 ± 11.3	0.12	−37.28 ± 2.67
DOPE/DOPA + rhodamine	164.8 ± 13.7	0.15	−35.94 ± 2.31

DOPA: 1,2-dioleoyl-sn-glycero-3-phosphate; DOTAP: 1,2-dioleoyl-3-trimethylammonium-propane.

**Table 2 ijms-24-01201-t002:** Encapsulation efficiency (EE%) of rhodamine in liposomal formulation (mean ± SD).

Liposome Composition	Encapsulation Efficiency (EE%)
DOPC/DOPE + rhodamine	25.3 ± 6.1
DOPC/DOTAP + rhodamine	29.9 ± 9.5
DOPE/DOPA + rhodamine	23.9 ± 8.7

**Table 3 ijms-24-01201-t003:** Means and standard deviation (SD) of total cells, live cells, dead cells and vitality percentage detected in samples of untreated Sertoli cells (SCs), SCs incubated with zwitterionic, anionic and cationic liposomes.

Samples	Vol.	Total Cells (n° × 10⁶)	Live (n° × 10⁶)	Dead (n° × 10^5^)	Vitality %
SCs	2 mL	3.17 ± 0.64	2.97 ± 0.55	1.93 ± 0.85	94.00 ± 0.02
SCs + Zwitterionic liposomes	2 mL	3.17 ± 0.47	3.00 ± 0.36	1.83 ± 0.83	94.00 ± 2.01
SCs + Anionic liposomes	2 mL	2.26 ± 0.60	2.10 ± 0.53	1.76 ± 0.710	93.00 ± 3.20
SCs + Cationic liposomes	2 mL	2.70 ± 0.10	2.46 ± 0.06	1.83 ± 0.40	93.00 ± 1.00

AMH and inhibin B, dosed in the medium of treated samples and controls (Table 4), did not show any significant difference.

**Table 4 ijms-24-01201-t004:** Means and standard deviation (SD) of Anti-Mullerian hormone (AMH) and Inhibin B in Sertoli cells alone (SCs), incubated with zwitterionic, anionic and cationic liposomes.

	SCs	SCs + Zwitterionic Liposomes	SCs + Anionic Liposomes	SCs + Cationic Liposomes
AMH	0.096 ± 0.056	0.77 ± 0.042	0.62 ± 0.08	0.66 ± 0.07
Inhibin B	21.41 ± 9.60	21.20 ± 7.54	23.8 ± 1.55	24.02 ± 3.38

## Data Availability

The data that support the findings of this study are available from the corresponding author upon reasonable request.

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
