# Peer review of "Effects and Mechanisms Activated by Treatment with Cationic, Anionic and Zwitterionic Liposomes on an In Vitro Model of Porcine Pre-Pubertal Sertoli Cells"

_ijms, 2023, doi:10.3390/ijms24021201_

Round 1

Reviewer 1 Report

The article describes the study of a method for delivering drugs to Sertoli cells using liposomes. The authors consider cationic, anionic, and zwitterionic liposomes and show that zwitterionic liposomes are the most promising compared to cationic/anionic ones, which cause rearrangement of the cytoskeleton and increase apoptosis. In general, this is an interesting study, but there are several points that raise questions.

In Table 4, AMH values ​​with cationic liposomes are half the standard deviation - the authors should check this point again. In this representation, the value of the standard deviation indicates that the value of AMH can be negative, which is impossible. In addition, for anionic liposomes there is also too much scatter in the data, judging by the standard deviation. And such a difference in standard deviation between groups does not allow for their correct comparison. Perhaps the authors should analyze their experimental data more carefully and/or increase the study sample.

When analyzing the structure of the cytoskeleton (lines 182-218), the authors presented data only for actin and vimentin, and concluded that no rearrangements of the cytoskeleton were observed under the action of zwitterionic liposomes. However, actin is a member of the family of microfilament proteins, vimentin is of intermediate filaments, but the authors did not analyze microtubules. At the same time, microtubules are one of the main structures of the cytoskeleton involved in cell division and intracellular transport. Accordingly, speaking about the possible use of liposomes as a means of drug delivery, it is necessary to analyze the state of the structure of microtubules.

Reviewer 2 Report

In the manuscript, ijms-2074948, authors aimed to understand the effects of treatment with cationic, anionic, and zwitterionic liposomes on Sertoli cells (SCs) to find a carrier of in vitro test drugs for SCs. 

I've found some minor aspects deserving clarification. 

It would be instructive if you could explain or contextualize in the manuscript the efficiency of encapsulation. 

SD values of AMH levels in SCS exposed to anionic and Cationic liposomes are too high. 

Could the authors clarify the reason why zwitterionic liposomes have less toxic effects on SCs compared with the others?

This can be considered a suitable drug delivery system, but I wonder how they are used under in vivo conditions. Could you please explain?

Round 2

Reviewer 1 Report

The authors corrected the AMH values and tried to describe why they did not analyze the state of microtubules. The authors responded to my suggestions and I think that the article can be accepted.